# Phosphates as Energy Sources to Expand Metabolic Networks

**DOI:** 10.3390/life9020043

**Published:** 2019-05-22

**Authors:** Tian Tian, Xin-Yi Chu, Yi Yang, Xuan Zhang, Ye-Mao Liu, Jun Gao, Bin-Guang Ma, Hong-Yu Zhang

**Affiliations:** 1Hubei Key Laboratory of Agricultural Bioinformatics, College of Informatics, Huazhong Agricultural University, Wuhan 430070, China; ttlym1989@163.com (T.T.); chuxy@webmail.hzau.edu.cn (X.-Y.C.); yyphoenix@163.com (Y.Y.); lym25517235@163.com (Y.-M.L.); gaojun@mail.hzau.edu.cn (J.G.); mbg@mail.hzau.edu.cn (B.-G.M.); 2Beijing National Center for Molecular Sciences, Institute of Theoretical and Computational Chemistry, College of Chemistry and Molecular Engineering, Peking University, Beijing 100871, China.; xuanz@pku.edu.cn

**Keywords:** origin of life, metabolism, phosphates, network expansion simulation, thermodynamic bottleneck, molecular clocks

## Abstract

Phosphates are essential for modern metabolisms. A recent study reported a phosphate-free metabolic network and suggested that thioesters, rather than phosphates, could alleviate thermodynamic bottlenecks of network expansion. As a result, it was considered that a phosphorus-independent metabolism could exist before the phosphate-based genetic coding system. To explore the origin of phosphorus-dependent metabolism, the present study constructs a protometabolic network that contains phosphates prebiotically available using computational systems biology approaches. It is found that some primitive phosphorylated intermediates could greatly alleviate thermodynamic bottlenecks of network expansion. Moreover, the phosphorus-dependent metabolic network exhibits several ancient features. Taken together, it is concluded that phosphates played a role as important as that of thioesters during the origin and evolution of metabolism. Both phosphorus and sulfur are speculated to be critical to the origin of life.

## 1. Introduction

Phosphates are basic components of many biomolecules and essential for modern biochemical reactions, but it is still not clear how phosphates play the critical role in metabolism in the origin of life. Phosphate minerals existing on the early Earth or in meteorites are thought to be the main sources of prebiotic phosphorus [1,2,3]. However, most of these phosphates are either insoluble in water or have low reactivity and thus they are considered to be problematic for primordial biological use [1].

To solve this ‘phosphorus problem’, Goldford and coworkers proposed a phosphorus-independent scenario for the emergence of protometabolism [4]. They constructed a phosphorus-independent protometabolism network starting from a set of prebiotically abundant compounds excluding phosphates. The obtained metabolic network contained various important metabolites and metabolic reactions, and exhibited the features of an ancient origin. Then, the researchers found that sulfur compounds (i.e., pantetheine) could alleviate the thermodynamic bottlenecks of the network expansion while phosphates (i.e., pyrophosphate or acetyl-phosphate) could not. Based on these findings, Goldford et al. proposed that a phosphorus-independent metabolism could exist before the emergence of the phosphate-based genetic coding system. However, this phosphorus-independent network could not produce nucleobases or ribose, which means that this network is unlikely a possible source of RNA.

What is more, phosphorus was thought to play a crucial role both in the prebiotic synthesis of important precursors of RNA and proteins [5,6,7] and in primordial energy metabolism [8,9,10]. These viewpoints support the importance of phosphorus in the origin of life. Moreover, the ‘phosphorus problem’ itself might have been solved by recent findings. Phosphite, a kind of water-soluble, reactive reduced-state phosphorus, was recently proposed to be an available prebiotic phosphorus source [10]. It has been proven that phosphite can be produced from the extraterrestrial phosphide mineral-schreibersite and is present in early Archean marine carbonates at significant levels [11,12]. This reduced-state phosphorus could generate orthophosphate, pyrophosphate and trimetaphosphate in plausible prebiotic environments [13]. With the presence of trimetaphosphate, diamidophosphate (DAP) could have formed and enable the phosphorylation and oligomerization of biologically meaningful molecules [14].

Goldford et al.’s new opinion stimulated our interest in the systematical exploration of the role that phosphorus played in the origin of metabolism. We noticed that Goldford et al. considered mere pyrophosphate and acetyl-phosphate but ignored other forms of primitive phosphates when investigating the thermodynamic bottlenecks of network expansion. This work started with alternative phosphates and adopted the network expansion algorithm [4,15,16,17] to simulate the expansion of metabolic networks. Then, the feasibility of this metabolic network on the early Earth was explored. Finally, biological features of this network were fully analyzed. This study reveals that: (i) some phosphorylated intermediates could efficiently alleviate thermodynamic bottlenecks; (ii) phosphorous-dependent metabolic network exhibits ancient biological features.

## 2. Materials and Methods

### 2.1. Data Sources

All KEGG reactions (i.e. chemical reactions in the KEGG database), compounds, the enzymes of reactions were downloaded from the KEGG database [18] (Release: 84.0, October 1, 2017). The putative LUCA genes (i.e. genes of the last universal common ancestor) were downloaded from the LUCApedia webpage [19]. Cofactor and PDB structures of the enzymes were downloaded from the Uniport database (Release: 2017_09). The folds and fold families of enzymes were downloaded from the SCOP database [20]. The architectures of enzymes were downloaded from the CATH database [21] (version 4.2). 

### 2.2. Reconstruction of the Background Metabolism Pool

KEGG Reactions that met the following conditions were removed: (i) reactions in which the molecular formulas of compounds were undefined; (ii) reactions that contained n-subunit polymers; (iii) reactions that contained the metabolites with “R” groups; (iv) reactions that were elementally imbalanced with exception of hydrogen. A stoichiometric matrix was then constructed using reaction equations. The final background metabolism pool consisted of 7376 reactions.

### 2.3. Network Expansion Simulation

The phosphorus-independent and -dependent metabolic networks were constructed by a network expansion algorithm which was first proposed by Ebenhöh et al. [15,16] and was described in detail in previous studies [4,15,16,17].

A seed set, namely, initial metabolite set, *M_0_* was first defined for the network expansion algorithm. Then, initial reaction set *R_0_* was constructed by identifying those reactions whose reactants were all included in *M_0_*. Reaction products *M_p_* of these reactions were further added into *M_0_* and subsequently a new metabolite set *M* (*M* = *M_0_* ∪ *M_p_*) was constructed. The new reactions *R_n_* whose reactants were present in *M* were identified and added into the reaction set *R* (*R*= *R_0_* ∪ *R_n_*). At each iteration *k*, the reaction set *R* and metabolite set *M* were updated with new added reactions and the products of these reactions (*R* = *R* ∪ *R_n_*, *M* = *M* ∪ *M_n_*). This process was terminated when no new reactions and products could be added into *M* and *R*.

The network expansion algorithm with different seed sets were executed. Every time, the initial seed set contained: (i) the possible abundant gases on the early Earth (dinitrogen, vapor water, hydrogen sulfide, and carbon dioxide); (ii) a possible prebiotic nitrogen source, i.e., ammonia; (iii) possible prebiotic carbon sources (acetate and formate); (iv) a possible prebiotic phosphorus source (i.e., orthophosphate, pyrophosphate and trimetaphosphate) (only for phosphorus-dependent metabolic network expansion simulation). The identification of the seed set was based on previous studies [4,9,13,22,23,24,25].

All reactions in this study came from the background metabolism pool. In our study, the reactions containing molecular oxygen were removed during the network expansion due to the anaerobic environment of the early Earth [17,26].

### 2.4. Thermodynamically Constrained Network Expansion Simulation

The thermodynamically constrained network expansion simulation was based on the network expansion algorithm. In the simulation, the cutoff value *τ* was set. The endergonic reactions in which the required free energies were above *τ* were removed. In other words, reactions with Δ*G_r_^0^* > τ were removed. The Δ*G_r_^0^* of the reactions was estimated using eQuilibrator [27,28]. It should be noted that there was a lack of free energy estimation in more than one third of KEGG reactions. Reactions with unknown free energies were assumed to be either all available or all unavailable. The results were very similar between these two treatments. In this paper, all reactions with unknown free energies were assumed as available. The rest steps in simulation were executed as they were in the network expansion algorithm.

To determine the potential thermodynamic bottlenecks of the network, the thermodynamically constrained network expansion algorithm was executed by adopting the same seed set used to construct the protometabolic network. To explore the influence of the phosphorylated intermediates from glycolysis, one of the prebiotically available intermediates (i.e., glucose 6-phosphate, glyceraldehyde 3-phosphate, glycerate 2-phosphate, glycerate 3-phosphate, and phosphoenolpyruvate) [29,30] was added into the seed set every time, and then the thermodynamically constrained network expansion algorithm was executed.

### 2.5. Scale-Limiting Reaction Detection

Reactions meeting the following conditions were defined as scale-limiting reactions: (i) the reactions triggering the dramatic expansion of networks; (ii) reactions whose removal will severely limit the expansion of networks.

In order to identify the network-scale-limiting reactions, the following steps were executed:(1).Obtaining the potential reactions: Reaction set *R_1_* and the corresponding metabolite set *M_1_* at the thermodynamic threshold *τ_1_* limiting the network expansion and reaction set *R_2_* and corresponding metabolite set *M_2_* at *τ_2_* (*τ_2_* = *τ_1_* + 1) were obtained(2).Identifying the reactions triggering the dramatic expansion of networks: At first, a metabolite *m* derived from difference set (*M_3_*) of *M_2_* and *M_1_* (*M_3_* = *M_2_* - *M_1_*) was added into the metabolite set *M_1_* to form a new metabolite set *M_4_* (*M_4_* = *M_1_* + *m*). Then, new networks at *τ_1_* were constructed with *M_4_*. Once the network expands dramatically, the corresponding reactions of *m* were considered to cause the dramatic expansion of networks. These steps were repeated for every metabolite in *M_3_*, and all of the expansion-triggering reactions were identified as the potential scale-limiting reactions.(3).Identifying the scale-limiting reactions: Since there may be more than one reaction limiting the dramatic expansion of the network, all combinations of potential scale-limiting reactions should be tested. Every combination of potential scale-limiting reactions was blocked in turn. The networks at the thermodynamic constraint *τ_2_* were constructed based on these abridged reaction sets. When dramatic network expansion was no longer observed, the reaction combinations were considered as scale-limiting ones. Thus, the final scale-limiting reactions were identified by analyzing all the combinations.

### 2.6. Protein Domain Age Estimation

In this study, the protein domains are classified according to SCOP and CATH protein structure classification schemes. The node distance (*nd*) values based on SCOP fold family (FF) were the united set of the *nd* values derived from several previous studies [31,32,33,34], the *nd* values based on CATH architecture (A) were derived from Bukhari et al.’s work [35].

The previous studies reported that the *nd* values of structural domains were closely related to their geological ages. Based on this finding, several molecular clocks were constructed [31,32,33,34]. In this study, the geological ages of proteins were defined according to the united set of those molecular clocks.

## 3. Results

### 3.1. Construction of Phosphorus-Dependent Metabolic Network

First, we attempted to evaluate the reliability of the network expansion simulation. First, we reconstructed the background metabolism pool with reactions and compounds from the updated version of the KEGG database [18]. The final updated background metabolism pool contained 7376 reactions (full-balanced network, Appendix A), and included 496 more reactions than the pool constructed by Goldford and coworkers [4]. Then, a phosphorus-independent metabolic network was reconstructed with updated background reaction data started with a pre-defined seed metabolite set. This seed set was the same as that defined by Goldford et al. and was composed of a set of prebiotically abundant compounds excluding phosphates. The final phosphorus-independent network included 329 reactions and 266 metabolites, containing almost all of the reactions of the network constructed by Goldford et al. (Appendix A). This result validated the reliability of network expansion simulation.

The phosphorus-dependent metabolic network was constructed with the same method except that prebiotic phosphates were added into the seed set. Due to the lack of KEGG reaction data of phosphite, we did not directly introduce phosphite as a phosphorus source. Phosphates we introduced here are orthophosphate, pyrophosphate and trimetaphosphate. These phosphates were widely thought to be present on the early Earth and could be prebiotically synthesized by phosphite [13,22,23,24,25]. Each time we introduced one of these phosphates and performed the network expansion simulation. The obtained phosphorus-dependent metabolic networks were composed of the same reactions (596 reactions) and metabolites (471 metabolites) (Figure 1 and Appendix A), implying that the network was robust to different phosphorus sources. 

The reactions and metabolites increased obviously in the phosphorus-dependent network. Although nucleobases were not found in the new-produced metabolites, ribose, which is also an essential component of RNA, was indeed produced in the phosphorus-dependent metabolic network, indicating the importance of phosphorus in the evolution of RNA synthesis.

### 3.2. Thermodynamic Bottleneck Alleviation by Primitive Phosphates

Thermodynamic constraints could limit the expansion of the metabolic network [36]. Phosphates play important roles in driving energetically uphill reactions. However, Goldford et al. claimed that phosphates such as pyrophosphate and acetyl-phosphate could not alleviate thermodynamic bottlenecks, while pantetheine could [4]. Their statement was supported by the updated simulation on the phosphorous-independent network (Figure 2A, Appendix A). Nevertheless, we are still wondering whether there exist any other forms of primitive phosphates that could serve as alternative alleviators for the thermodynamic bottleneck?

Glycolysis-like reactions could spontaneously occur in a plausible ancient marine environment [37]. Many phosphorous intermediates of glycolysis, including glucose 6-phosphate, glyceraldehyde 3-phosphate, glycerate 2-phosphate, glycerate 3-phosphate, and phosphoenolpyruvate, were speculated to be prebiotically synthesized [29,30]. All of these phosphorous intermediates are present in the phosphorus-dependent metabolic network. Thus, we attempt to explore whether the glycolysis-generated metabolites can alleviate thermodynamic bottlenecks and can promote the expansion of the early metabolic network.

The expansion of phosphorus-dependent metabolic network was re-simulated under the thermodynamic constraints. During the simulation, endergonic reactions in which the required free energies were above a cutoff value, *τ*, were blocked during the expansion of the network. When *τ* was below 51 kJ/mol, the scale of the network was strictly limited with reactions and metabolites limited to <26 and <30, respectively (Appendix A). When *τ* exceeded this threshold, the network expanded dramatically (Figure 2A, Appendix A). It seemed impossible for early metabolism to overcome the energetic constraint of 51 kJ/mol because endergonic reactions with Δ*G_r_^0^* (standard transformed Gibbs energies) above 30 kJ/mol needed to be activated by exergonic reaction like ATP hydrolysis [36]. However, this kind of exergonic reaction might be unavailable in the primitive world [36].

Then, the glycolysis-derived phosphorylated intermediates were introduced into the network during the thermodynamically constrained network expansion. With the addition of phosphorylated intermediates, the bottlenecks limiting the network expansion were reduced to below 30 kJ/mol (Figure 2B–F, Appendix A), implying the expansion of these networks is thermodynamically feasible without other energy sources [36]. At the end of each simulation, the thermodynamically constrained networks contained at least 338 metabolites and 413 reactions.

To exclude the influence of sulfur, we removed hydrogen sulfide from the seed set and found that its removal had little impact on the thermodynamically constrained network expansion (Figure 2B–F, Appendix A), suggesting that sulfur made no significant contribution to alleviating the thermodynamic bottlenecks of the phosphorus-dependent network expansion.

The reactions involved in the dramatic expansion of the metabolic networks were also investigated. The dramatic expansions of the networks disappeared when certain reactions (i.e., R00024, R01070 and R00346) were blocked, indicating that these reactions played a critical role in limiting the expansion of the networks (Figure 2 and Appendix A). Then, the feasibility of these scale-limiting reactions at the early stage of evolutionary history of metabolism was explored. Reversible reaction R00024 was observed to be the most common scale-limiting reaction in five thermodynamically constrained networks with different phosphorous intermediates. In R00024, glycerate 3-phosphate and ribulose 1,5-bisphosphate were key metabolites. This reaction is catalyzed by RubisCO (D-ribulose 1,5-bisphosphate carboxylase/oxygenase, EC: 4.1.1.39), which was assumed to originate 3.5 Gy ago [38,39]. Besides, it was reported that RubisCO catalyzed this reaction by offering COO^-^, and H^+^ [40,41]. All these ions could exist in the primitive Earth environment, which suggested that reaction R00024 might occur before RubisCO appeared. 

Reaction R01070 is catalyzed by beta-D-fructose-1,6-bisphosphate D-glyceraldehyde-3-phosphate-lyase (EC: 4.1.2.13) [42]. Inferred by protein-structure-based molecule clocks [31,32,33,34], this enzyme catalyzing appeared earlier than 3 Gy. However, the feasibility of R00346 in ancient time remains unknown due to the lack of knowledge of its structure of catalyzing enzyme (oxaloacetate carboxy-lyase, EC: 4.1.1.38). Taken together, it can be concluded that phosphorylated intermediates could have alleviated the thermodynamic bottlenecks of metabolic network expansion, at least the expansion constrained by R00024 and R01070.

### 3.3. Ancient Origin of Phosphorus-Dependent Metabolic Network

The ancient origin of the phosphorus-dependent metabolic network was evaluated by the biological characteristics of the network. We analyzed enzymes of the phosphorus-dependent network to explore the potential biological features associated with ancient metabolism. The phosphorus-dependent network was found to be enriched with the enzymes, orthologs and protein fold families of LUCA (*p* < 10^−2^, Fisher’s exact test, Figure 3A). This result implied that a great portion of the reactions in the phosphorus-dependent network existed in the early life. This network was also enriched with the enzymes which contained metal cofactors Mg^2+^, Zn^2+^ and FeS (*p* < 0.05, Fisher’s exact test, Figure 3B). The relative higher requirement for metal ions might be a remnant of prebiotic catalysts, because existent enzymes may still retain characters of prebiotic catalysts, such as usage of metal cofactors [43,44]. Moreover, pyridoxal phosphate was considered to play a critical role in prebiotic transamination [45]. The enzymes using pyridoxal phosphate as a cofactor were also enriched in the network (*p* < 10^−5^, Fisher’s exact test, Figure 3B).

Besides, the structure of proteins was rather conserved during evolution and could serves as molecular fossils in the study of the early history of biochemistry evolution [31]. In previous research, molecular clocks based on different protein structure classification schemes (i.e., SCOP and CATH) were established and the relative ages of protein domains were characterized by node distances (*nd*) [31,32,33,34,35]. Node distance is the distance from the position of a taxon of protein domain structures on the phylogenetic tree to the root node, with the scale from 0 (most ancient) to 1 (most recent). It has been shown that *nd* values of protein domain structures correlate strongly with their geological times. In this study, the ages of enzymes in the phosphorus-dependent and -independent networks were inferred using these molecular clocks at both fold family level (SCOP classification) [31,32,33,34] and architecture level [35]. The accumulation patterns of the enzyme ages in two networks exhibited no significant difference (Kolmogorov–Smirnov test, *p* > 0.05, Figure 3C), suggesting that enzymes in phosphorus-independent network are not more ancient than phosphorus-dependent counterpart.

Taken together, the above results indicated that both phosphorus-dependent and phosphorus-independent networks are at least as ancient as LUCA. Besides, the phosphorus-dependent network retains a higher requirement for metal ions and pyridoxal phosphate, which might be remnants of prebiotic chemistry.

## 4. Discussion

It is undeniable that phosphorus is essential for modern metabolism from both material and energy perspectives. Phosphates are basic components of important biomolecules and play an important role in energy transduction, signal transmission and redox catalysis. Considering its critical role in metabolism, phosphorus is thought to make great contributions to the origin of life. To examine the role of phosphorus in the origin of metabolism, we constructed a metabolic network using a network expansion algorithm. The phosphorus-dependent metabolic network contains much more metabolites than the phosphorus-independent counterpart. Among the phosphorus-dependent network, ribose is produced, which is an essential component of RNA, indicating the significance of phosphorus for the primordial synthesis of RNA.

To explore the influence of phosphorus on the thermodynamic feasibility of ancient metabolic system, the thermodynamically constrained network expansion with various forms of phosphates was simulated. This study found that some phosphorous intermediates of the glycolytic pathway could dramatically alleviate the thermodynamic bottlenecks and promote the expansion of the network. Further study of scale-limiting reactions (i.e., R00024 and R01070) during the thermodynamically constrained network expansion showed that the expansion of ancient metabolic network might be feasible with the presence of phosphorous intermediates such as glucose 6-phosphate and glyceraldehyde 3-phosphate. 

The biological characteristics of phosphorus-dependent network were comprehensively analyzed, and results showed that this network exhibits several ancient features. The enzymes in this network were enriched with LUCA elements and metal-based cofactors which were considered to be used in original biochemical reactions [4,44]. The ages of enzymes in phosphorus-dependent and -independent networks exhibited similar accumulation patterns. These results indicated that both phosphorus-dependent and phosphorus-independent networks are at least as ancient as LUCA. Moreover, phosphorus-dependent network exhibits some more ‘primitive’ traits, such as retaining a relative higher requirement for metal ions and pyridoxal phosphate.

In summary, our research demonstrates that (i) some high-energy phosphates can ensure the primitive metabolism under feasible energetic constraints; (ii) the phosphorous-dependent metabolism might originate in the very early stage of biochemical processes. 

Therefore, it can be speculated that phosphates are as important as thioesters for the origin and evolution of metabolism. Both phosphorus and sulfur are critical to the origin of life on Earth. This has meaningful implications for extraterrestrial life detection. Recently, Enceladus, a satellite of Saturn, was reported to have a global liquid water ocean and the jets from this ocean contain simple organic chemicals, suggesting that Enceladus provides some basic conditions to fulfill the existence of life [46]. However, phosphorus and sulfur have not yet been detected in the ocean jets of Enceladus [46], that casts a shadow over the existence of Enceladus life.

It should bear in mind that this work is based on KEGG reactions. The premise of KEGG reactions is that there must be a cellular environment. Thus, the prebiotic reactions which might have been replaced during the evolution of life cannot be included in the current networks. As a consequence, the conclusion of this study may have some limitations. Besides, the phosphorus-dependent network does not produce nucleobases, which implied that there still is a gap to evolve RNA. Finally, why phosphorous intermediates of glycolysis mechanism could alleviate the thermodynamic bottlenecks remains to be elucidated, in particular considering the fact that addition of these triose phosphates may cause complex changes of metabolism.

## Figures and Tables

**Figure 1 life-09-00043-f001:**
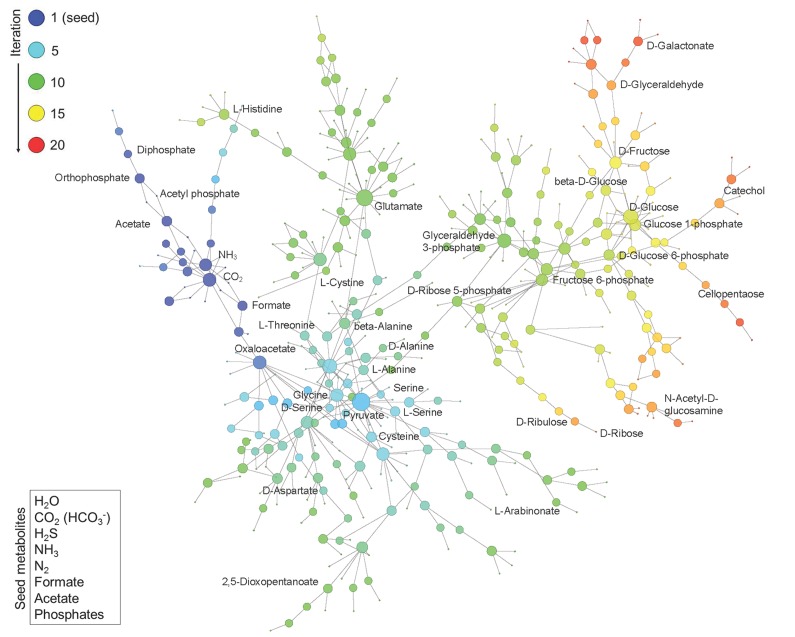
Construction of Phosphorus-Dependent Metabolic Network. Network expansion simulation was executed using a set of defined seed compounds (bottom left box) and all balanced reactions in the background metabolism pool derived from the updated KEGG reactions. The figure displays the obtained phosphorus-dependent network in which metabolites are linked if they have a reactant-product relationship during the expansion. The metabolites generated at different iteration steps during the network expansion process are represented by nodes in different colors. The size of node represents the degree of the node, i.e., the number of reactions added in the subsequent iteration.

**Figure 2 life-09-00043-f002:**
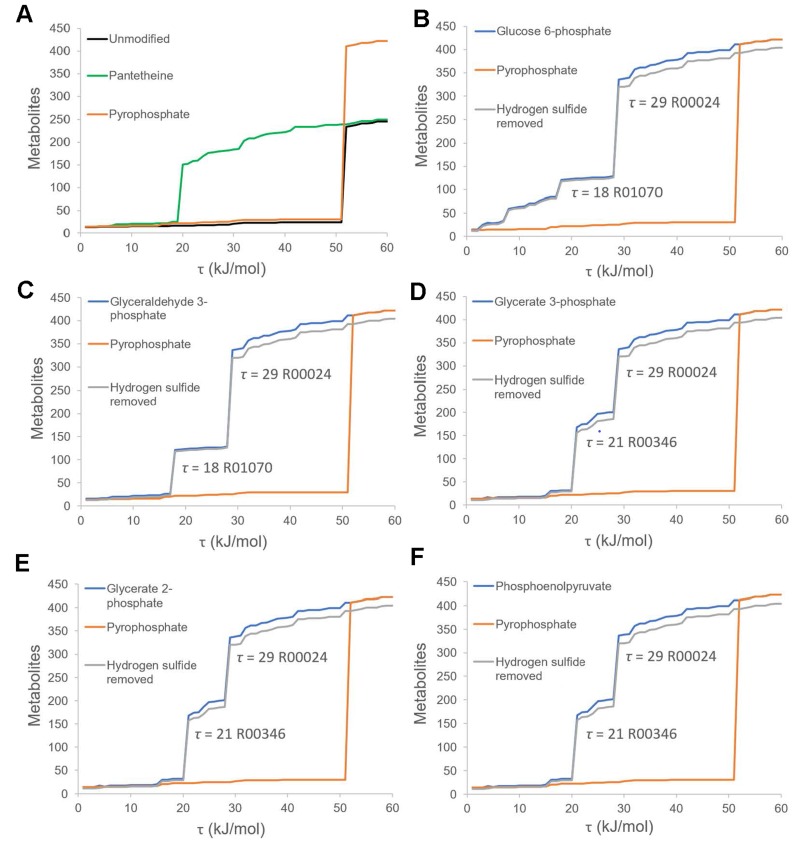
Thermodynamically Constrained Network Expansion. Thermodynamically constrained network expansion was simulated by using different seed sets. Endergonic reactions with Δ*G_r_^0^* exceeding a thermodynamic threshold *τ* were defined as impossible. For each value of *τ* (*x* axis), we plotted the size of the final expanded network in terms of the number of metabolites (*y* axis). (**A**) displays the comparison of the network sizes of the unmodified phosphorus-dependent network (black line), the pyrophosphate-coupled network (with the addition of pyrophosphate in the seed set) (orange line), and the pantetheine-coupled network (with the addition of pantetheine in the seed set) (green line) at different thermodynamic thresholds, *τ*. (**B**–**F**) display the comparison of the network sizes of the pyrophosphate-coupled network (orange line), the phosphorylated intermediates-coupled network (blue line), and the phosphorylated intermediates-coupled network without hydrogen sulfide (gray line) at different thermodynamic thresholds, *τ*. The thermodynamic bottlenecks and the reactions limiting the scale of different phosphorylated intermediates-coupled networks are shown in corresponding figures. The used phosphorylated intermediates include: glucose 6-phosphate (B), glyceraldehyde 3-phosphate (C), glycerate 2-phosphate (D), glycerate 3-phosphate (E), phosphoenolpyruvate (F).

**Figure 3 life-09-00043-f003:**
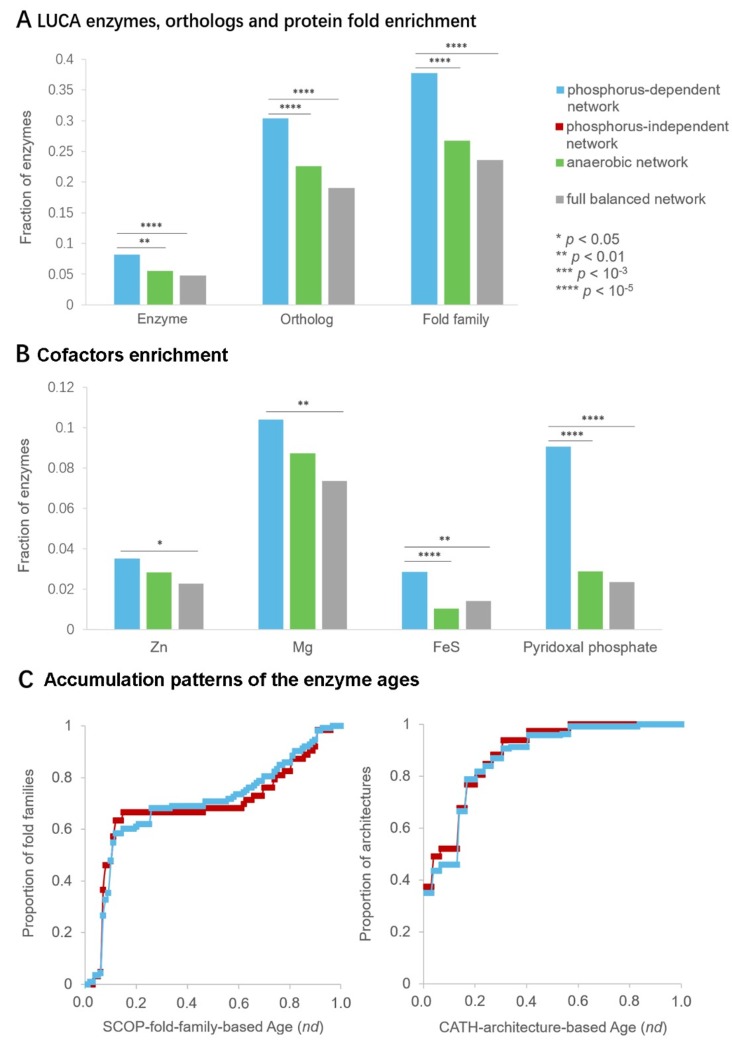
Biological Characteristics of Phosphorus-Dependent Metabolic Network. (**A**) The phosphorus-dependent network is enriched with enzymes, ortholog genes, and protein fold families that are thought to be present in LUCA, relative to all metabolic reactions in background metabolism pool (full-balanced network) or to the oxygen-independent (anaerobic) portion of the full network. (**B**) The phosphorus-dependent network is enriched with metal cofactors (Zn^2+^, Mg^2+^ and FeS) and pyridoxal phosphate, relative to all metabolic reactions in background metabolism pool or to the oxygen-independent portion of the full network. (**C**) The accumulation patterns of the enzyme ages in two networks show no significant difference. All of these results show the ancient biological characteristics of phosphorus-dependent metabolic network, suggesting that both phosphorus-dependent network and phosphorus-independent network are at least as ancient as LUCA. The significance was analyzed by Fisher’s exact test or Kolmogorov–Smirnov test: * *p* < 0.05; ** *p* < 0.01; *** *p* < 10^−3^; **** *p* < 10^−5^.

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
