# Peer review of "Phosphates as Energy Sources to Expand Metabolic Networks"

_life, 2019, doi:10.3390/life9020043_

Reviewer 1 Report

This is a very nicely performed and presented piece of science which reports the results of network expansion simulations into the potential role of prebiotically available phosphorus sources in metabolic evolution.

Overall, the work is solid and deserved of publication in my view. There are however, some important points that need to be raised in relation to the context of this work which any reader would need to be aware of in advance as they are not as yet incorporated into the paper.

As I understand it, the network expansion modelling relies on manipulation of the KEGG database of metabolic reactions. Hence, everything that could possibly accrue from a manipulation of the kind reported here can only have any relevance to a living system which is itself capable of providing an environment which supports such reactions; ie: the premise of KEGG reactions is that there must be a cellular environment in which these networks evolve. This in turn means that life must already have emerged to the point at which their is a genotype-phenotype relationship with concomitant ribosomal machinery and suitable cytoplasmic environment on which to perform the reactions. In other words, life must have already begun for any such simulation to have value. The protein enzymes that catalyse the reaction in the metabolic simulation may be classed as primitive but even then such molecular clock dating based on protein folding can only be necessity only go back as far as protein folding having known function (indeed the referenced paper by Wang, #26) only traces such protein ages back to the great oxidation event which gave rise to aerobic life, not to the emergence of life itself or the prebiotic milieu that preceded it.

So, what this means is that any conclusions drawn from the simulations reported here cannot have any relevance to prebiotic chemistry per se as life needs to have produced cellular machines for the modelling to have any validity. This really needs to be made clear to the reader as there are clear potential problems in using a top-down approach to make assertions of bottom-up evolution.

Author Response

Point 1: As I understand it, the network expansion modelling relies on manipulation of the KEGG database of metabolic reactions. Hence, everything that could possibly accrue from a manipulation of the kind reported here can only have any relevance to a living system which is itself capable of providing an environment which supports such reactions; ie: the premise of KEGG reactions is that there must be a cellular environment in which these networks evolve. This in turn means that life must already have emerged to the point at which their is a genotype-phenotype relationship with concomitant ribosomal machinery and suitable cytoplasmic environment on which to perform the reactions. In other words, life must have already begun for any such simulation to have value.

Response 1: Thanks for the thoughtful comments. Start from the unanticipated opinion that phosphorus may not be important to the origin of life, the theme of this article is to explore the roles of phosphorus played in the expansion of ancient metabolic system. The KEGG reactions-based analysis showed that phosphorus-dependent network might be as ancient as phosphorus-independent network. The existence of these two networks might not could be traced as far back as the prebiotic stage. Results of our work showed that as least part of these two networks could be existed at the early stage of biochemical process. Addition of speculated prebiotic phosphates could cause the alleviation of thermodynamic bottlenecks of network expansion, implies that primitive phosphates could facilitate the expansion of metabolic networks, thus provide the possibility of the ancient phosphorus-dependent metabolism on the early Earth. In the revised manuscript, we corrected our conclusions (line 61-63, line 252-254, line 350-352, line 361-363).

Point 2: The protein enzymes that catalyse the reaction in the metabolic simulation may be classed as primitive but even then such molecular clock dating based on protein folding can only be necessity only go back as far as protein folding having known function (indeed the referenced paper by Wang, #26) only traces such protein ages back to the great oxidation event which gave rise to aerobic life, not to the emergence of life itself or the prebiotic milieu that preceded it.

Response 2: We agree that molecular clock may not be an appropriate method to trace prebiotic biochemistry. The molecular clock used here was to prove that phosphorus-independent network is not more ancient than phosphorus-dependent counterpart. In the revised manuscript, we rewrote this part (line 199 to line 254). We firstly point out that catalysts of early metabolic reactions could be short peptides, metal cofactors, minerals or clays, and would much different from they are today. However, enzymes, which gradually replace the prebiotic catalysts and catalyze modern metabolic reactions may still retain characters of ancient catalysts [1,2]. Then, the ancient origin of the phosphorus-dependent metabolic network was evaluated by the biological/chemical characteristics of the network from three aspects. 1) The enzymes in this network were enriched with LUCA elements and metal-based cofactors which were considered to be used in original biochemical reaction; 2) The ages of enzymes in phosphorus-dependent and -independent networks exhibited similar accumulation patterns; 3) The metabolites in these two networks were relatively smaller and had a stronger molecular polarity and higher water solubility, compared to the metabolites in the modern metabolic network, which is in accordance with the concept that life came from ocean. Taken together, the above results indicated that the birth of phosphorus-dependent network was not later than that of phosphorus-independent network, and both networks may have ancient origins.

References

1.       Goldford J.E., Hartman H., Smith T.F., Segrè D. Remnants of an ancient metabolism without phosphate. Cell 2017, 168, 1126-1134.

2.       Sousa F.L., Martin W.F. Biochemical fossils of the ancient transition from geoenergetics to bioenergetics in prokaryotic one carbon compound metabolism. Biochim. Biophys. Acta 2014, 1837, 964–981.

Point 3: What this means is that any conclusions drawn from the simulations reported here cannot have any relevance to prebiotic chemistry per se as life needs to have produced cellular machines for the modelling to have any validity. This really needs to be made clear to the reader as there are clear potential problems in using a top-down approach to make assertions of bottom-up evolution.

Response 3: Our purpose was to explore whether phosphorus is as important as sulfur to the origin of metabolism. Although methods and data used may be inappropriate to trace the prebiotic chemistry, results of this work indeed support the importance of phosphorus to the ancient metabolism and imply that both phosphorus-dependent network and phosphorus-independent network might have ancient origins. In the revised manuscript, we made corresponding corrections to our conclusions (line 61-63, line 252-254, line 350-352, line 361-363) and indicated limits of this work (line 371-377).

Reviewer 2 Report

I think this is an interesting paper, providing a useful counterbalance to some ideas on the role of phosphorus in the pre-biotic and early biotic chemistry that lead to life. The role of phosphorus in that chemistry remains highly contentious, and I personally think we are nowhere near resolving this issue (or any other in the OOL field). But that is not to say that we should not try. This paper makes some good arguments for why phosphorus could be incorporated into the very earliest chemical reactions of life.

The network expansion algorithm is well known. Whether it represents a plausible evolutionary path is of course contentious. Modern metabolic networks can go through very convoluted pathways, reflecting the limitations in chemical mechanisms available to life and the contingencies of evolution. Thus a network expansion algorithm how modern metabolism (i.e LUCA metabolism) evolved if it evolved without intermediate steps that were later abandoned (for example, if the original synthesis of adenine was via condensation of cyanide, but synthesis through an ATP-dependent route was a more efficient use of carbon or energy and hence the cyanide route was replaced in LIUCA, and hence in all modern organisms). The authors should be more explicit about what models of expansion based on modern metabolism can tell us, what it may tell us, and what is highly speculative that it tells us.

They should also be more explicit about what stage of chemistry they are discussing. Gene-coded enzyme-catalysed metabolism was at the last stage of the emergence of life. Abiotic chemical reactions are the first. What stage in the process does the author’s analysis apply to? I did not get a clear picture of this. Discussion R01070 they talk about uncatalysed reactions, and yet there is a large section on the evolution of proteins. The authors should be clearer about whether they are talking about prebiotic chemistry, the earliest ‘biotic’ chemistry (catalysed by inefficient, relatively non-specific catalysts, which may have been proteins or may not), or an evolved, coded metabolism.

The enzyme dating study I find unconvincing. It is technically well done, but it does not tell us anything about prebiotic metabolism. All it shows is that both phosphorus-dependent and phosphorus-independent subnetworks were present a very long time ago, probably in LUCA. Although this is not stated explicitly, the non-phosphate network must be a subset of the phosphate network (unless you can make things with metabolism lacking phosphorus that you cannot make in the same metabolism that contains phosphorus, which seems wrong). So it would be odd if the phosphate network were older than a substantial subset of the phosphate network (although if it were, this would strongly hint that phosphate metabolism was original). If the phosphate network were significantly younger than the non-phosphate network, that would imply that life evolved a fully functional metabolic and genetic system without RNA or DNA, which would be surprizing to say the least. So, inevitably, their proteins are the same age.

Thus the statement “The origin history of phosphorus-dependent and -independent metabolism are traced by  molecular clocks of protein domain structures. The similar accumulation patterns of the enzyme ages of phosphorus-dependent and phosphorus-independent networks suggested that both networks originated in the same stage” (lines 300-303) is not correct. What this shows is that the origin of the ancestors of the modern proteins that catalyse these reactions is not resolvable using the techniques used. It does not show when they originated, it does not show that these were descended from the initial catalysts that catalysed these networks (such catalysts could be RNAs, clays etc, and not proteins at all), and and does not say anything about prebiotic chemistry. The authors should reconsider this part of the paper, and either explain why evolved, post-LUCA enzyme structures are relevant to pre-enzymatic proto-biological chemistry, or remove this part of the argument.

The calculations on thermodynamic limitations on network expansion are more convincing. Addition of pyrophosphate to the network does not allow its expansion, addition of other phosphorylated species does. I confess to finding the result that pyrophosphate does not allow network expansion to be odd. Surely

Adenosine + PP -> AMP + P

AMP + PP -> ADP + P

ADP + P -> ATP

And from ATP you can drive any reaction in metabolism, but this may be a naïve understanding. It would help in this paper if the authors added a few words to explain why in their networks including phosphorus you cannot get from PP to ATP. I would also ask whether addition of sugar phosphates was in fact allowing the driving of redox metabolism, which is the real driver of diversity. Adding triose phosphates would allow the reduction of FeS proteins through glycolysis, and reduced FeS proteins (or NADH/FADH2 equivalents) are key to many reactions. If you add phosphate but remove FeS proteins, what happens? (This may be a complex and extensive piece of work to do, but the authors should at least acknowledge that adding triose phosphate is a complex change in metabolism, and is not just a way of adding phosphorus.

The big problem with this approach is the assumption that molecules such as triose phosphates could be made in any realistic prebiotic scenario. There is an extensive literature saying they could, and an equally extensive literature saying that they could not (or at least, could not be made in any usable amounts, and if made would be rapidly broken down again). I would want the authers to say that it has been *speculated* that such compounds can be made under prebiotic conditions, not state it as if it were established. (making them in the lab with pure reagents under defined conditions etc is *not* the same as prebiotic synthesis!)

Smaller points.

R01070 obviously does have an energy barrier to C-C bond cleavage, otherwise fructose-1,6-biphosphate would not be stable in solution, which it is. That is not to say that R01070 could not occur on prebiotic Earth, but it would need catalysis. The para on R01070 should be rewritten, or removed.

I would not use ‘primarily’ to mean ‘first’. ‘Primarily’ has implications of ‘most important’, whereas I think you just mean ‘this is the first step’.

Well done for discussion Enceladus, but a) find a better reference than a NASA press release (eg https://science.sciencemag.org/content/sci/356/6334/155.full.pdf), and b) the Cassini fly-through of the Enceladus plumes did not detect S or P, but this is *as expected* - NH3 was present in 0.4 – 1.3%: if P and S are present at their cosmological ratio to N, then S would be present 0.2 – 0.8% (barely detectable) and P would be present at 0.002% - 0.01%, which would be well below the limits of detection. Absence of evidence is not evidence of absence! Phosphorus might be present on Enceladus. SO at least soften this to say “… has not yet been detected…”.

Author Response

Point 1: The network expansion algorithm is well known. Whether it represents a plausible evolutionary path is of course contentious. Modern metabolic networks can go through very convoluted pathways, reflecting the limitations in chemical mechanisms available to life and the contingencies of evolution. Thus a network expansion algorithm how modern metabolism (i.e LUCA metabolism) evolved if it evolved without intermediate steps that were later abandoned (for example, if the original synthesis of adenine was via condensation of cyanide, but synthesis through an ATP-dependent route was a more efficient use of carbon or energy and hence the cyanide route was replaced in LIUCA, and hence in all modern organisms). The authors should be more explicit about what models of expansion based on modern metabolism can tell us, what it may tell us, and what is highly speculative that it tells us.

They should also be more explicit about what stage of chemistry they are discussing. Gene-coded enzyme-catalysed metabolism was at the last stage of the emergence of life. Abiotic chemical reactions are the first. What stage in the process does the author’s analysis apply to? I did not get a clear picture of this. Discussion R01070 they talk about uncatalysed reactions, and yet there is a large section on the evolution of proteins. The authors should be clearer about whether they are talking about prebiotic chemistry, the earliest ‘biotic’ chemistry (catalysed by inefficient, relatively non-specific catalysts, which may have been proteins or may not), or an evolved, coded metabolism.

Response 1: We appreciate the thorough reading that you have made on our manuscript and the numerous suggestions for revision. Start from the unanticipated opinion that phosphorus may not be important to the origin of life, the theme of this article is to explore the roles of phosphorus played in the expansion of ancient metabolic system. Results of our work showed that phosphorus-dependent network might be as ancient as phosphorus-independent network, part of these two networks could be existed at the early stage of biochemical process. Discussion about uncatalyzed reactions was to explore the possibility of the ancient phosphorus-dependent metabolism on the early Earth. Corresponding conclusions were corrected in the revised manuscript (line 61-63, line 252-254, line 350-352, line 361-363). In the rewritten part, we pointed out that what this study can tell us, i.e phosphorous-dependent metabolic network shows critical ancient biological/chemical features and some phosphorylated intermediates could efficiently alleviate thermodynamic bottlenecks (line 62-64); what it may tell us, i.e the phosphorous-dependent metabolism might originate in very early stage of biochemical processes (line 362-363), phosphates are as important as thioesters for the origin and evolution of metabolism. Both phosphorus and sulfur are critical to the origin of life on Earth (line 364-365).

Point 2: The enzyme dating study I find unconvincing. It is technically well done, but it does not tell us anything about prebiotic metabolism. All it shows is that both phosphorus-dependent and phosphorus-independent subnetworks were present a very long time ago, probably in LUCA. Although this is not stated explicitly, the non-phosphate network must be a subset of the phosphate network (unless you can make things with metabolism lacking phosphorus that you cannot make in the same metabolism that contains phosphorus, which seems wrong). So it would be odd if the phosphate network were older than a substantial subset of the phosphate network (although if it were, this would strongly hint that phosphate metabolism was original). If the phosphate network were significantly younger than the non-phosphate network, that would imply that life evolved a fully functional metabolic and genetic system without RNA or DNA, which would be surprizing to say the least. So, inevitably, their proteins are the same age.

Thus the statement “The origin history of phosphorus-dependent and -independent metabolism are traced by molecular clocks of protein domain structures. The similar accumulation patterns of the enzyme ages of phosphorus-dependent and phosphorus-independent networks suggested that both networks originated in the same stage” (lines 300-303) is not correct. What this shows is that the origin of the ancestors of the modern proteins that catalyse these reactions is not resolvable using the techniques used. It does not show when they originated, it does not show that these were descended from the initial catalysts that catalysed these networks (such catalysts could be RNAs, clays etc, and not proteins at all), and and does not say anything about prebiotic chemistry. The authors should reconsider this part of the paper, and either explain why evolved, post-LUCA enzyme structures are relevant to pre-enzymatic proto-biological chemistry, or remove this part of the argument.

Response 2: In the revised manuscript, we rewrote this part (line 199-254). We firstly point out that catalysts of early metabolic reactions could be short peptides, metal cofactors, minerals or clays, and would much different from they are today. However, enzymes, which gradually replace the prebiotic catalysts and catalyze modern metabolic reactions may still retain characters of ancient catalysts [1,2]. Then, the ancient origin of the phosphorus-dependent metabolic network was evaluated by the biological/chemical characteristics of the network from three aspects. 1) The enzymes in this network were enriched with LUCA elements and metal-based cofactors which were considered to be used in original biochemical reaction; 2) The ages of enzymes in phosphorus-dependent and -independent networks exhibited similar accumulation patterns (The molecular clock used here was to prove that phosphorus-independent network is not more ancient than phosphorus-dependent counterpart); 3) The metabolites in these two networks were relatively smaller and had a stronger molecular polarity and higher water solubility, compared to the metabolites in the modern metabolic network, which is in accordance with the concept that life came from ocean. Taken together, the above results indicated that the birth of phosphorus-dependent network was not later than that of phosphorus-independent network, and both networks may have ancient origins.

References

1.       Goldford J.E., Hartman H., Smith T.F., Segrè D. Remnants of an ancient metabolism without phosphate. Cell 2017, 168, 1126-1134.

2.       Sousa F.L., Martin W.F. Biochemical fossils of the ancient transition from geoenergetics to bioenergetics in prokaryotic one carbon compound metabolism. Biochim. Biophys. Acta 2014, 1837, 964–981.

Point 3: The calculations on thermodynamic limitations on network expansion are more convincing. Addition of pyrophosphate to the network does not allow its expansion, addition of other phosphorylated species does. I confess to finding the result that pyrophosphate does not allow network expansion to be odd. Surely

Adenosine + PP -> AMP + P

AMP + PP -> ADP + P

ADP + P -> ATP

And from ATP you can drive any reaction in metabolism, but this may be a naïve understanding. It would help in this paper if the authors added a few words to explain why in their networks including phosphorus you cannot get from PP to ATP.

Response 3: The resulted phosphorous-dependent metabolic network did not contain adenosine, that is why we cannot get ATP from PP in this network.

Point 4: I would also ask whether addition of sugar phosphates was in fact allowing the driving of redox metabolism, which is the real driver of diversity. Adding triose phosphates would allow the reduction of FeS proteins through glycolysis, and reduced FeS proteins (or NADH/FADH2 equivalents) are key to many reactions. If you add phosphate but remove FeS proteins, what happens? (This may be a complex and extensive piece of work to do, but the authors should at least acknowledge that adding triose phosphate is a complex change in metabolism, and is not just a way of adding phosphorus.

Response 4: Thanks for the thoughtful comments. In the revised manuscript, we acknowledged that that addition of triose phosphates may cause complex changes of metabolism (line 376-377).

Point 5: The big problem with this approach is the assumption that molecules such as triose phosphates could be made in any realistic prebiotic scenario. There is an extensive literature saying they could, and an equally extensive literature saying that they could not (or at least, could not be made in any usable amounts, and if made would be rapidly broken down again). I would want the authers to say that it has been *speculated* that such compounds can be made under prebiotic conditions, not state it as if it were established. (making them in the lab with pure reagents under defined conditions etc is *not* the same as prebiotic synthesis!)

Response 5: Thank you for your suggestion, we had revised it in the manuscript (line 284).

Point 6: R01070 obviously does have an energy barrier to C-C bond cleavage, otherwise fructose-1,6-biphosphate would not be stable in solution, which it is. That is not to say that R01070 could not occur on prebiotic Earth, but it would need catalysis. The para on R01070 should be rewritten, or removed.

Response 6: The paragraph on R01070 was rewritten and the quantum chemical calculation part of this reaction was removed in the revised manuscript (line 329-331).

Point 7: I would not use ‘primarily’ to mean ‘first’. ‘Primarily’ has implications of ‘most important’, whereas I think you just mean ‘this is the first step’.

Response 7: We are sorry for the inaccurate use of the words “primarily”, we had revised it in the manuscript (line 170).

Point 8: Well done for discussion Enceladus, but a) find a better reference than a NASA press release (eg https://science.sciencemag.org/content/sci/356/6334/155.full.pdf), and b) the Cassini fly-through of the Enceladus plumes did not detect S or P, but this is *as expected* - NH3 was present in 0.4 – 1.3%: if P and S are present at their cosmological ratio to N, then S would be present 0.2 – 0.8% (barely detectable) and P would be present at 0.002% - 0.01%, which would be well below the limits of detection. Absence of evidence is not evidence of absence! Phosphorus might be present on Enceladus. SO at least soften this to say “… has not yet been detected…”.

Response 8: Thank you for your suggestion, we had revised the corresponding part of the manuscript (line 369-370).

Round  2

Reviewer 1 Report

I’m happy now that the authors have worked to clarify the constraints that naturally accrue from a database study such as performed here. What the authors have demonstrated here has clear potential in the field of metabolic evolution once primitive life had begun. Any potential relevance to prebiotic systems is far less clearly evident

Author Response

Point 1: I’m happy now that the authors have worked to clarify the constraints that naturally accrue from a database study such as performed here. What the authors have demonstrated here has clear potential in the field of metabolic evolution once primitive life had begun. Any potential relevance to prebiotic systems is far less clearly evident.

Response 1: We appreciate the thorough reading that you have made on our manuscript and your thoughtful comments.

Reviewer 2 Report

I think this has much good material in it, and should be published. However I think the argumentation distracts the reader rather than helping them at the moment. Specifically, I think that your main argument is that Goldford’s arguments as to why phosphate cannot have been a key component of primitive metabolism and thioesters were is wrong, for connectivity and thermodynamic grounds if you assume sources of organophosphate molecules to primitive metabolism. This is *backed up* by evolutionary analysis, but this is supporting evidence.

Re-ordering the paper to bring this logic out will help.

Details:-

I remain completely unconvinced that you can deduce prebiotic chemistry from modern sequence and fold information. As another review said, and I agree and said in similar terms, that tells you what modern organisms do, and with a *lot* of assumptions tells you what LUCA did. It tells you nothing about prebiotic chemistry.

Thus lines 204-5: "However, enzymes, which gradually replace the prebiotic catalysts and catalyze modern metabolic reactions may still retain characters of ancient catalysts [4,38]” Well, sure, they “may”. The modern genes *may* retain sequences from prebiotic RNA. But it is highly contentious, and you do not want to be basing your hypothesis on something that is itself a speculation.

It is astonishing that a PP-based pathway is said not to make adenosine, but that is what Goldford et al say – their Table S2 does not list any purines! That is absurd – how did primitive life evolve the ability to make RNA without being able to make purines? The problem is that the purine synthesis pathway requires ATP, so without ATP you cannot make ATP. I am surprized the authors did not pick up on this. But I cannot find adenosine (or adenine) in their supplementary table of phosphate independent or dependent networks either. If their network cannot make RNA, then obviously it cannot evolve into something that can make proteins by translation or have genes, and so those prebiotic catalysts cannot ‘evolve into’ LUCA enzymes. They have to address this!

 “Taken together, the above results indicated that the birth of phosphorus-dependent network was not later than that of phosphorus-independent network, and both networks may have ancient origins.” No, it shows that they were both present as far back as the techniques can probe. If I carbon date dinosaur fossils and trilobite fossils, I could say that  the ‘birth’ of the dinosaurs was not later than the birth of the  trilobites. But this is an incorrect inference. All I have done is say that the origin of dinosaurs and trilobites pre-dates any date that 14-C dating can provide. . Similarly, all the authors have done is show that phosphorus-dependent biochemistry and phosphorus-independent chemistry predates their dating system, which *at the very best* only probes the metabolism of LUCA.

It is not known that life originated in the ocean. It originated in an environment containing water, but it must also have been a dehydrating environment to drive condensation chemistry. And  even if it did, it does not follow that early metabolites are hydrophilic. How are membranes made? How do proteins fold if all the amino acids are hydrophilic? And anyway, of course phosphorus-containing metabolites are more hydrophilic – they contain phosphate groups! You could make the same argument and say that the earliest metabolites were all acids, because acids are more hydrophilic than average for metabolites. This new section is not correct, and should be removed.

I think this is a decent argument in the following terms:

·         Goldford et al say that you can build a primitive, pre-RNA ‘metabolism’ without phosphorus, but cannot build one with phosphorus that alleviates key thermodynamic bottlenecks. And it does not make adenosine, which means it cannot evolve RNA.

·         We show that you can, if you assume that phosphorus is present in forms other than just as PP.

·         This also alleviates thermodynamic bottlenecks in the network which the non-P or pyrophosphate-only approach does not alleviate.

·         So including organo-phosphates probably happened very early in the evolution of metabolism

·         Supporting this, molecular evolution evidence is that phosphate-dependent chemistry is at least as ancient as LUCA, and shows more ‘primitive’ traits such as retaining a higher requirement for metal ions (which one might suppose is a remnant of prebiotic chemistry).

I think if you re-order the argument in that way, and make less of the evolutionary analysis as a key point in your argument (which it is not), then it will be fine.

Smaller point: What is the molecule on the left of Figure 4?  My guess is that it is the gem diol reaction intermediate, in which case it would be nice to say so. So what is the energy barrier to make this intermediate (remembering that gem diols are inherently thermodynamically disfavoured in water)? You say R00024 could have occulted abiologically, but does it? If you put ribulose-1,6-diplhosphate in an acid carbonate buffer, does it spontaneously fission? I am pretty sure it does not, no matter how many quantum calculations you do to prove otherwise, but I may be wrong. You could do that experiment!

I think if the authors view their protein structure work as supporting their basic idea rather than evidence for it, bring the network analysis to the fore (as that is what the paper is about!), and mention the huge limitations of the networks (no adenosine!) and what that implies, it will be fine, and a useful counter-argument to the Goldford et al paper.

Author Response

Point 1: Thus lines 204-5: "However, enzymes, which gradually replace the prebiotic catalysts and catalyze modern metabolic reactions may still retain characters of ancient catalysts [4,38]” Well, sure, they “may”. The modern genes *may* retain sequences from prebiotic RNA. But it is highly contentious, and you do not want to be basing your hypothesis on something that is itself a speculation.

Response 1: Thank you for your suggestion, we had rewrote the corresponding part of the manuscript (line 260-271).

Point 2: It is astonishing that a PP-based pathway is said not to make adenosine, but that is what Goldford et al say – their Table S2 does not list any purines! That is absurd – how did primitive life evolve the ability to make RNA without being able to make purines? The problem is that the purine synthesis pathway requires ATP, so without ATP you cannot make ATP. I am surprized the authors did not pick up on this. But I cannot find adenosine (or adenine) in their supplementary table of phosphate independent or dependent networks either. If their network cannot make RNA, then obviously it cannot evolve into something that can make proteins by translation or have genes, and so those prebiotic catalysts cannot ‘evolve into’ LUCA enzymes. They have to address this!

Response 2: Actually, phosphorus-independent network of Goldford et al could not produce any nucleobases (including adenosine) or ribose. Although nucleobases were not found in the phosphate-dependent metabolic network, ribose, which is also an essential component of RNA, was indeed produced in this network, indicating the importance of phosphorus in the evolution of RNA synthesis. In the revised manuscript, we added this result and its corresponding conclusion (line 42-44, line 176-179, line 311-314).

Point 3: “Taken together, the above results indicated that the birth of phosphorus-dependent network was not later than that of phosphorus-independent network, and both networks may have ancient origins.” No, it shows that they were both present as far back as the techniques can probe. If I carbon date dinosaur fossils and trilobite fossils, I could say that the ‘birth’ of the dinosaurs was not later than the birth of the trilobites. But this is an incorrect inference. All I have done is say that the origin of dinosaurs and trilobites pre-dates any date that 14-C dating can provide. Similarly, all the authors have done is show that phosphorus-dependent biochemistry and phosphorus-independent chemistry predates their dating system, which *at the very best* only probes the metabolism of LUCA.

Response 3: Thank you for your suggestion, we had revised it in the manuscript (line 300-303).

Point 4: It is not known that life originated in the ocean. It originated in an environment containing water, but it must also have been a dehydrating environment to drive condensation chemistry. And even if it did, it does not follow that early metabolites are hydrophilic. How are membranes made? How do proteins fold if all the amino acids are hydrophilic? And anyway, of course phosphorus-containing metabolites are more hydrophilic – they contain phosphate groups! You could make the same argument and say that the earliest metabolites were all acids, because acids are more hydrophilic than average for metabolites. This new section is not correct, and should be removed.

Response 4: Thank you for your suggestion, we had removed this section in the revised manuscript.

Point 5: I think this is a decent argument in the following terms:

Goldford et al say that you can build a primitive, pre-RNA ‘metabolism’ without phosphorus, but cannot build one with phosphorus that alleviates key thermodynamic bottlenecks. And it does not make adenosine, which means it cannot evolve RNA.

We show that you can, if you assume that phosphorus is present in forms other than just as PP.

This also alleviates thermodynamic bottlenecks in the network which the non-P or pyrophosphate-only approach does not alleviate.

So including organo-phosphates probably happened very early in the evolution of metabolism

Supporting this, molecular evolution evidence is that phosphate-dependent chemistry is at least as ancient as LUCA, and shows more ‘primitive’ traits such as retaining a higher requirement for metal ions (which one might suppose is a remnant of prebiotic chemistry).

I think if you re-order the argument in that way, and make less of the evolutionary analysis as a key point in your argument (which it is not), then it will be fine.

Response 5: Thanks for your thoughtful comments. We had re-ordered our arguments in the revised manuscript.

Point 6: What is the molecule on the left of Figure 4? My guess is that it is the gem diol reaction intermediate, in which case it would be nice to say so. So what is the energy barrier to make this intermediate (remembering that gem diols are inherently thermodynamically disfavoured in water)? You say R00024 could have occulted abiologically, but does it? If you put ribulose-1,6-diplhosphate in an acid carbonate buffer, does it spontaneously fission? I am pretty sure it does not, no matter how many quantum calculations you do to prove otherwise, but I may be wrong. You could do that experiment!

Response 6: Because the lack of experimental evidence, this controversial part was removed from the revised manuscript.

Point 7: I think if the authors view their protein structure work as supporting their basic idea rather than evidence for it, bring the network analysis to the fore (as that is what the paper is about!), and mention the huge limitations of the networks (no adenosine!) and what that implies, it will be fine, and a useful counter-argument to the Goldford et al paper.

Response 7: Thank you for your suggestion. In the revised manuscript, we had acknowledged the huge limitation of the networks, that is, the phosphorus-dependent network does not produce nucleobases (including adenosine), which implied that there still is a gap to evolve RNA (line 344-346).